# Smart Operation of Climatic Systems in a Greenhouse

Aurora González-Vidal [1,2], José Mendoza-Bernal [1], Alfonso P. Ramallo [1,*], Miguel Ángel Zamora [1], Vicente Martínez [3] and Antonio F. Skarmeta [1]

1 Department of Information and Communication Engineering, University of Murcia, 30100 Murcia, Spain
2 Information Technologies Institute, The Centre for Research & Technology, Hellas (ITI-CERTH), 57001 Thessaloniki, Greece
3 Department of Vegetal Nutrition, Centro de Edafología y Biología Aplicada del Segura del Consejo Superior de Investigaciones Científicas (CEBAS-CSIC), 30100 Murcia, Spain
* Correspondence: alfonsop.ramallo@um.es

**Abstract:** The purpose of our work is to leverage the use of artificial intelligence for the emergence of smart greenhouses. Greenhouse agriculture is a sustainable solution for food crises and therefore data-based decision-support mechanisms are needed to optimally use them. Our study anticipates how the combination of climatic systems will affect the temperature and humidity of the greenhouse. More specifically, our methodology anticipates if a set-point will be reached in a given time by a combination of climatic systems and estimates the humidity at that time. We performed exhaustive data analytics processing that includes the interpolation of missing values and data augmentation, and tested several classification and regression algorithms. Our method can predict with a 90% accuracy if, under current conditions, a combination of climatic systems will reach a fixed temperature set-point, and it is also able to estimate the humidity with a 2.83% CVRMSE. We integrated our methodology on a three-layer holistic IoT platform that is able to collect, fuse and analyze real data in a seamless way.

**Keywords:** smart agriculture; greenhouse technologies; artificial intelligence

## 1. Introduction

Traditional agriculture can no longer adequately absorb the world's growing demand for food. Water scarcity and climate change seriously threaten global food security for present and future generations. In this context, smart agriculture emerges as a new approach that focuses on optimizing production efficiency, increasing quality, minimizing environmental impact and reducing the use of resources (energy, water and fertilizers) [1].

The emergence of smart agriculture implies high productivity ratios thanks to the modernization of irrigation and climate systems and the application of advanced control systems. Greenhouses are widespread within Europe, with an area of nearly 405,000 dedicated hectares [2], and the total greenhouse agricultural area in Spain exceeded 73 thousand hectares in 2021 [3]. In addition, awareness of the potential environmental impact of these intensive systems and, above all, the existence of more sustainable methods and technologies in agriculture is permeating into the population, which causes changes in behaviors and new regulations for farmers and technology providers.

Among the advances applied in protected crops with strong growth potential is the use of soil-less crops, which allow for increases in productivity and a more efficient use of water and fertilizers. Initially, most soil-less farming systems were open, where drainage is released into the environment. This leads to a high water consumption, and approximately 31% of nitrates and 48% of potassium applied during the crop cycle are discharged into the environment, causing the contamination of aquifers and possibly environmental problems of eutrophication also [4]. However, in recent years, European policies have been aimed at making agricultural practices more sustainable, with initiatives such as the Green Deal [5]. In the Mediterranean area, where more than 60% of its production is under greenhouses,

the implementation of closed systems in crops without soil would have a significant environmental impact, since it would notably reduce discharges of nitrates and phosphates into the environment. In addition, it would mean significant savings in fertilizers and water, these being scarce resources in the Mediterranean area.

An important challenge in smart agriculture applied to greenhouses is the high consumption of energy when one wants to carry out a certain degree of automation of the production system. The use of renewable energies and energy efficiency applied to the different climatic and fertilization machinery makes these new agricultural systems more profitable than traditional agriculture and reduces their carbon footprint. This is because the solar resource tends to be high in agricultural sites.

The introduction of artificial intelligence (AI) and massive data processing (big data) as tools in the field of smart agriculture and greenhouses opens up a range of possibilities. These new techniques help to address the challenges of this new type of agriculture, such as the optimization of production with respect to resources, pest control, rapid reaction to adverse situations, predictions of extreme events, etc., in order to make this type of agriculture a sustainable and profitable industrial sector in the long term.

In this work, we present an ICT platform that is composed of three different technological layers. The solution has the latest innovations, such as the use of big data processing and machine learning as a service applied to a closed-cycle hydroponic greenhouse in order to estimate the effects of choosing a specific combination of climatic systems. The authors believe that this is a relevant research question as the piping of the hydroponic system and the fact that the roots are not embedded in the soil contribute to substantial inertia, making the thermodynamics of the environment affecting the plant truly different than those of a conventional greenhouse.

The specific objectives of this work are as follows:

- To create a ICT platform that is able to gather data, fuse it, analyze it and provide greenhouse-related services;
- To create a methodology that anticipates the thermal response of a greenhouse according to real-time variables using artificial intelligence:
  - Gather and prepare the data to analyze the operation of different climatic systems and their effect on temperature and humidity;
  - Create classification models that anticipate if a combination of climatic systems will reach the desired temperature in a greenhouse in a certain time;
  - Create regression models to estimate greenhouse humidity;
- To integrate the created methodology into the created platform.

This will serve as a tool to optimize the use of different climatic systems attending the energy consumption in order to increase the efficiency of the installation.

This paper is organized as follows: Section 2 contains the review of the literature on the use of AI in greenhouses. Section 3 provides an overview of the ICT system architecture (explaining the three layers) and Section 4 delves into the analytical proposal, including the data processing and the description of the classification and regression models. Section 5 shows the experimental set-up and how our methodology can be integrated into any platform, and discusses the novelty and originality together with some practical applications of the work. Section 6 concludes the work.

## 2. State of the Art

Agriculture has a primary role in meeting global nutrition needs. In recent decades, the food and agriculture sectors have passed through substantial changes [6]. Improvements in agricultural productivity are expected to be increased in order to ensure planned development, improvements and maintenance that may cover the needs of growing world populations [7]. The excessive use of phytosanitary products, soil tillage techniques and production must necessarily change if we want to meet current challenges. Climate change, the depletion of fossil resources, environmental issues (protection of biodiversity, fight

against pollution, etc.) and health standards require a change in model to one that tends towards a more sustainable use of the soil [8].

Protected cropping is defined as growing within, under or sheltered by structures such as cover material, shade cloths, plastic tunnels or greenhouses. The level of protection and control can range from an inexpensive canopy (e.g., fabric/cloth) in a field to greenhouses or complete controlled environment horticulture systems [9]. In recent years, soil-less cultivation has become increasingly important as a promising strategy for growing a variety of crops. It is difficult to cultivate crops in horticulture greenhouses under hot and arid climate conditions. The main challenge is to provide a suitable greenhouse indoor environment with sufficiently low costs and with low environmental impacts. Once in place, greenhouses are highly non-linear, complex, multi-input and multi-output (MIMO) systems when looked at as thermodynamic enclosures that respond to changes in morphology and operations with changes in temperature and humidity [10]. Sustainable agriculture cultivation in greenhouses is constantly evolving thanks to new technologies and methodologies able to improve the crop yield and solve the common concerns that occur in protected environments [11]. Due to climate change, variations in weather have occurred in the last decades, and the introduction of new methodologies and technologies has become essential in supporting agriculture and optimizing greenhouse productions [12].

Soil-less cultivation (hydroponics) allows for a higher control of plant growth thanks to the better control of the feeding medium. This leads to a high productivity and better product quality, as well as a very high efficiency of water and fertilizer use [13]. Especially in regions with limited water or soil resources, hydroponic cultivation techniques can open up new approaches to food production. Not all systems are equally efficient, nor can they be applied in all areas and locations. Nowadays, the cultivation of horticultural crops, including leafy and fruiting vegetables and medicinal herbs with pharmaceutical value, are commercially grown in recycled (i.e., recirculating) hydroponics under controlled environments [14]. Recirculating hydroponics has great implications in practice under controlled environment agriculture toward economic considerations and environmental sustainability. Even more controlled agriculture is generally used in indoor farming plant factories for producing a range of high-value crops such as leafy and fruiting vegetables and medicinal plants under artificial light organized vertically [15]. In addition to the lighting and nutrient supply, the numerous systems, applications, substrates, organisms and their economic viability and sustainability have to be considered in order to obtain an understanding of whether and when it is worthwhile to use soil-less techniques [16].

Hydroponics could solve a considerable weakness of Mediterranean greenhouses. This is the large amount of energy required to maintain optimal environmental conditions for crop growth [17], which limits the operation period to approximately 9 months due to very high summer temperatures. The major challenge in Mediterranean greenhouses is to find ways to improve the yield per drop of water and unit of energy. An increased investment is required and needs to be considered in terms of the return on investments [18]. Greenhouses and indoor farming systems play an important role in providing fresh food, such as fruits and vegetables being high in vitamins and minerals. Controlled-environment greenhouses combine a high crop production per unit area with a high water use efficiency per unit of product [19], but at the cost of a high energy demand [20] and high investments. Increasing agricultural systems' resource efficiency is a key action for producing adequate food quantities in semi-arid Mediterranean regions while coping with water scarcity, environmental constraints and economic issues. There have been several studies on the high consumption of energy that some greenhouse solutions may have [21]. Recently, Soussi et al. (2022) provided an updated literature review of the climate control methods and cooling systems, with a particular focus on their reliability under hot and arid climate conditions. The main criteria of the performance evaluation are the effectiveness of these systems in generating suitable climate conditions for crops, and their reduction in the use of energy and water. Due to their lightweight construction and inefficient insulation, greenhouses are considered as one of the most energy-intensive agents of the agricultural

industry. According to the literature, they consume more fossil-fuel-generated energy than that used by buildings of a similar size [22]. Smart agriculture applied to greenhouses is considerably interesting from an energy point of view. The growing request for the quality and production of fruits and vegetables imposes on the producer the equipping of greenhouses with suitable heating and cooling units to regulate the temperature, humidity and lighting, driving a healthy and controlled production. In smart applications, in general, it is mandatory to select proper high-efficiency technical solutions to reduce the energy consumption and, as a consequence, the production costs and environmental impact [12].

The future smart agriculture will be based on advanced data acquisition combined with several technologies. All of these outlooks involve data management in a certain way. With the development of open-source and big data, different techniques have emerged to remedy the limitations of traditional decision-making systems [23]. Sustainable agricultural development is a significant solution with fast population development through the use of information and communication (ICT) in precision agriculture, and has produced new methods for making cultivation more productive, proficient and well-regulated while not contributing more to climate change. Big data (machine learning, deep learning, etc.) is among the vital technologies of ICT, employed in smart agriculture for their rather large data analytic capabilities to abstract significant information and to assist agricultural professionals in comprehending full-farming practices on all of their facets, helping them to take precise decisions [24].

## 3. System Architecture

The future of smart agriculture will be based on advanced data acquisition combined with several technologies. All of these outlooks involve data management in a certain way. With the development of open-source and big data, different techniques have emerged to remedy the limitations of traditional decision-making systems [23]. Sustainable agricultural development is a significant solution to fast population development through the use of information and communication (ICT) in precision agriculture, which has produced new methods for making cultivation more productive, proficient and well-regulated while not contributing more to climate change. Big data (machine learning, deep learning, etc.) is among the vital technologies of ICT, employed in smart agriculture for their rather large data analytic capabilities to abstract significant information and to assist agricultural professionals in comprehending full-farming practices on all of their facets, helping to take precise decisions [24].

The data sources of the lower layers are fed into an NGSI-LD-based platform (an evolution from the European FIWARE initiative, based on the ETSI standards institute), enabling massive data management and providing support for the integration of machine learning and data analytic tools. This third layer (services layer) allocates the database with all of the information about the status of the crop and parameters, and it represents the interface with users. Moreover, the FIWARE-based platform employs standardized IoT-data protocols to facilitate the acquisition, integration and exchange of massive data from CPS and gateways, and also provides an interface to the big data techniques. The overall architecture scheme can be seen in Figure 1.

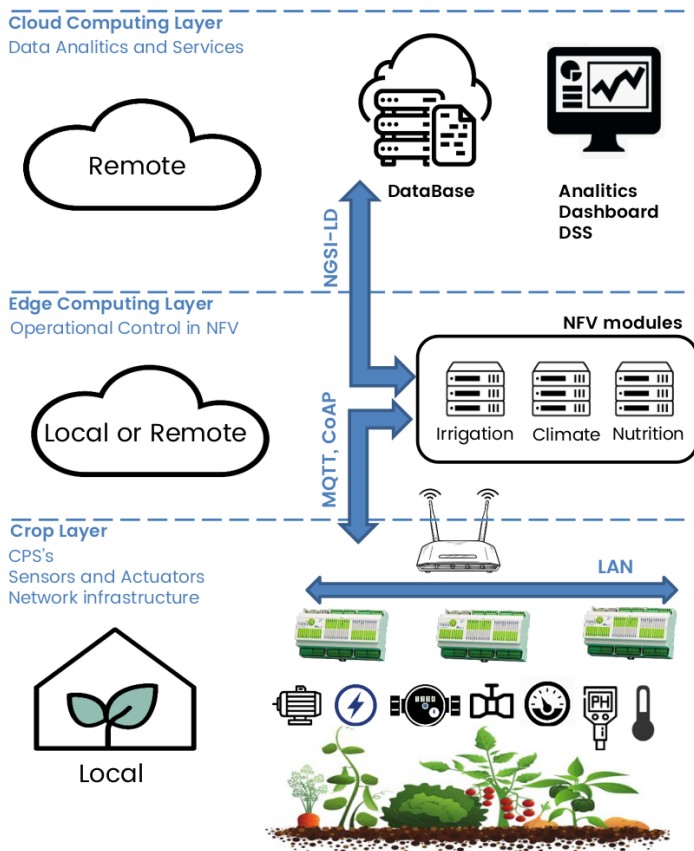

**Figure 1.** Architecture of the control system in the greenhouse.

## 4. Materials and Methods

The greenhouse under study had a set of equipment mainly deployed inside a machinery room beside the crops within the greenhouse. These facilities were composed of several hardware units; the climate unit is the one that this work focuses on.

The greenhouse had an IPv6 over Low Power Wireless Personal Area Network (6LoW-PAN) deployment, where sensors are directly interconnected using Internet protocols. The sensors measure temperature, humidity, solar radiation and $CO_2$. A CPS device was provided with peripherals dedicated to managing the different climatic devices. Although the platform is capable of managing up to thirteen climatic systems, the physical facilities of the prototype greenhouse have the following hardware available:

- Ventilation system with four overhead motorized windows.
- Thermal-shade screen system on the roof, with an electro-mechanical traction system for opening and closing. It receives 48% of shadow and 55% of energy saving.
- Air cooling system, which consists of a humidifier module and three helical extractors. The extractors have a flow of 38,000 $m^3$/h.
- Air fog system used to humidify and cool the greenhouse through water evaporation. It has a pressure pump of 1 HP and 25 L, and an air compressor of 4 HP and 50 L.

### 4.1. Data Preprocessing

The greenhouse sensors provide climatological data on temperature, humidity, radiation and photosynthetically active radiation (PAR). The latter discriminates the spectral range of solar radiation from 400 to 700 nanometers, which photosynthetic organisms are able to use in the process of photosynthesis. The data were collected between 1st February and 31st August, including, therefore, the hottest period in Murcia, Spain, when the true effect of the climate systems in lowering the temperature and controlling the humidity in the desired ranges can be seen. For some variables, such as temperature and humidity,

there were two sensors placed at 1.5 m (edge of the plants) and 3 m (above the plants) and places within the greenhouse. For analytical purposes, we used the average value for both.

While the sampling frequency of the sensors was 60 s, the climate systems can be switched on or off for less time, and also their sampling may not coincide in time. In order to homogenize the databases, we upsampled the time series to 10 s. For this purpose, a simple linear interpolation was performed by taking the values provided by the sensors of two consecutive samples ($v[i]$ and $v[i+1]$), dividing the time interval between both measurements into equal parts and assigning to each of them the value:

$$v(k) = v[i] + \frac{v[i+1] - v[i]}{6} \cdot k, \qquad \forall k \in \{1, \dots, 6\}, \tag{1}$$

where $i = \{1, \dots, N-1\}$.

$$v'(6 \cdot i + k) = v[i] + \frac{v[i+1] - v[i]}{6} \cdot k \tag{2}$$

where $i = \{0, \dots, N-1\}$ and $k = \{0, \dots, 6\}$.

Categorical variables were stored only every time a change in their status occurs. For example, windows return their percentage of opening and climatic systems, for which, data are available for the date and time of switching on and off; they were replicated in a similar way to the numerical variables, but taking into account that the intermediate values to be assigned between two consecutive measurements $w[i]$ and $w[j]$ are:

$$w'(k) = w[i], \qquad \forall k \in \{i+1, \dots, j-1\} \tag{3}$$

In addition to the sensors inside the greenhouse, we integrated the data from an external weather station that is next to the greenhouse. The weather station provides data on temperature, relative humidity, radiation, wind direction, wind speed and precipitation. The sampling frequency of the sensors in the station was five minutes. The same linear interpolation was applied for 10 s of upsampling to homogenize this data also.

The so-called vapor pressure deficit (vpd) was calculated inside and outside the greenhouse. The vapor-pressure deficit is the difference (deficit) between the amount of moisture in the air and how much moisture the air can hold when it is saturated. This variable is closely related to crop development and yield [25,26], and could be useful for estimating which climate systems should be switched on at which time. The vpd was calculated as follows in Equation (4).

$$vpd(t, rh) = svp(t) - \frac{rh}{100 \cdot svp(t)} \tag{4}$$

where $t$ is the temperature in Celsius degrees (ºC), $rh$ is the relative humidity (percentage) and $svp(t)$ is the saturation vapor pressure (kPa), and is calculated by the Tetens formula [27] that can be seen in Equation (5).

$$svp(t) = 0.6108 \cdot e^{\frac{17.27 \cdot t}{t + 237.3}} \tag{5}$$

Once the data were generated, we looked for missing values, errors in the sensor measurements or inconsistencies. There was a period of time (between 8 June and 23 July) when one of the humidity sensors in the greenhouse did not work properly. To compensate for these small gaps, two regression models were built as a function of the other variables and taking into consideration the climate systems. The regression models were radial support vector machines (RSVM) and a random forest (RF). These two models were chosen because they are well known and perform well in regression problems. The rest of the variables were within reasonable parameters and no missing values were found.

The regression models were trained and evaluated with the data, where the humidity inside the greenhouse was known. A total of 255,745 samples were used for training and

63,936 observations for testing, which is an 80–20 setup. To evaluate the performance of the models during training, 10 repeats of five-fold cross-validation were performed. The final values used for the RSVM model were $\sigma$ = 0.12 and C = 2.27, and the final values used for the RF model were min.node.size = 2, mtry = 6 and splitrule = "extratrees". The test results are shown in Table 1:

**Table 1.** Combinations of climate systems, number of occurrences observed and average switch-on times.

| Combination | Occurrences | Avg. Time in Seconds |
|---|---|---|
| Two recirculators | 2469 | 65 |
| One fan and two recirculators | 52 | 26 |
| Two fans | 83 | 28 |
| Two fans and two recirculators | 2407 | 56 |
| Two fans, two recirculators and cooling system | 49 | 28 |
| Three fans and two recirculators | 2438 | 45 |
| Three fans, two recirculators and cooling system | 1257 | 5050 |

RF is better than RSVM in all metrics. The coefficients of determination $R^2$ are close to 1, so the models fit the data perfectly. As the two models were trained and evaluated on the same datasets, statistical inference can be performed to determine if there are differences between using one or the other. Using Student's *t*-test method, the differences between the two models were found to be significant at 95% confidence (*p*-value adjustment by Bonferroni's method = 0.12 > $\alpha$ = 0.05). Therefore, the RF model was chosen to estimate the missing points of the humidity sensor.

For each combination of climate systems, we created a different table that includes the values of the above variables every 10 s.

*4.2. Climatic Systems Combinations*

Among the systems in place to modify the thermal conditions inside the greenhouse are: three fans, two recirculators, a ventilation system with four motorized upper windows, a thermal-shade screen system and the air cooling system. The ventilation system and the thermal screen system on the roof were used less than 1% of the time during this study, so the analysis of variables showed that these variables have a variance close to zero and were discarded. Therefore, only the three fans, the two recirculators and the air cooling system were used to modify the greenhouse thermal conditions.

Of the systems that were actually used to modify the conditions inside the greenhouse, seven combinations of them were found. Table 2 shows the combinations of climate systems that were found during this study, the number of occurrences of each of them and the average time of switching on.

**Table 2.** Test accuracy results of the four models for the different combinations of climate systems.

| Climatic Systems | Model | Accuracy |
|---|---|---|
| 2 fans + recirculators | Random forest | 89.2% |
| | Linear SVM | 74.5% |
| | Radial SVM | 81.9% |
| | Logistic regressor | 78.7% |
| 3 fans + recirculators | Random forest | 89.9% |
| | Linear SVM | 75.4% |
| | Radial SVM | 84.8% |
| | Logistic regressor | 79.4% |
| 3 fans + recirculators + cooling | Random forest | 98.4% |
| | Linear SVM | 88.8% |
| | Radial SVM | 86.5% |
| | Logistic regressor | 88.8% |

The statistical analysis carried out for each combination of climate systems found shows that the average switch-on times differ from one system to another, as can be seen in Table 2.

Considering that the two recirculators alone were used to homogenize the greenhouse conditions and were switched on periodically, this combination of systems was not taken into account when deciding which systems should be switched on to achieve the ideal climatic conditions inside the greenhouse. The statistical analysis showed that this combination of systems was only able to reduce the temperature slightly when the outside temperature decreased.

The statistical analysis showed that there were three combinations of climate systems (one fan and two recirculators, two fans, and two fans, two recirculators and the cooling system) that were switched on together very few times during this study. The average switch-on times of these systems were less than 30 s, and they did not change the temperature inside the greenhouse in any case. Therefore, these systems were discarded due to insufficient quality information.

In summary, three combinations of systems were used to decrease the greenhouse temperature: (1) two fans and two recirculators, (2) three fans and two recirculators and (3) three fans, two recirculators and a cooling system.

*4.3. Methodology: Binary Classification for Climatic Systems and Regression for Humidity Prediction*

In this work, we created a methodology that consists of the sequential application of two algorithms. The first one will output a logical value that indicates if a certain combination of climatic systems under the current conditions would be able to reach the desired temperature in the desired time, and it was formulated as a binary classification model. The second one will output the final humidity reached when using such a combination of climatic systems under the current conditions after the established time. These models are going to be trained using a machine learning (ML) approach as in the data interpolation.

ML is a part of the AI paradigm that studies the mathematical algorithms that can learn patterns using examples or previously collected data and then make a prediction or classification over new data [28].

For the first part, we are examining only binary classification, $Y = 0$ being the target variable when the required temperature is not reached using a particular combination of climatic systems and $Y = 1$ when it is reached. Several inputs were used related to the weather outside the greenhouse and meteorological conditions within the greenhouse. This is a form of supervised learning because the training and testing datasets contain a response label and the algorithm observes the input vector and attempts to learn a probability distribution to predict $y$ given $x$ [29]. The objective of the algorithm is to learn a

set of weights on a subset of the data that minimizes the error or loss between the ground truth and predicted value in order to precisely classify the input to the associated label [30].

Logistic regression and support vector machines (SVMs) are popular binary classification algorithms and a random forest is also very well-known in ML practices. We chose these models, including linear and radial kernels for SVMs for our purpose.

With a similar approach but from a regression point of view, we used the same strategy, including the same algorithms since they were valid for both classification and regression, for predicting the humidity.

### 4.4. Manual Data Augmentation

The data collected from the greenhouse could have been seen as insufficient in obtaining robust classification models. Additionally, the data were unbalanced after being partitioned between train/validation and test sets, as can be seen in Figure 2, so the models were biased toward the majority classes. Due to this, a manual data augmentation process was applied to the training data of the three combinations of climate systems that were successfully lowering the temperature inside the greenhouse. This data augmentation trained the models with balanced data and helped to better predict new unseen examples (increases in the generalizability of the models). For example, from an observation where the target temperature was reached by turning on certain climatic systems for $S$ seconds, we can generate new observations with the same conditions as before, but, with turn-on times shorter than $S$ seconds, we found that those climatic systems could not reach the target. This process generates new examples by applying common sense and adds value to the data.

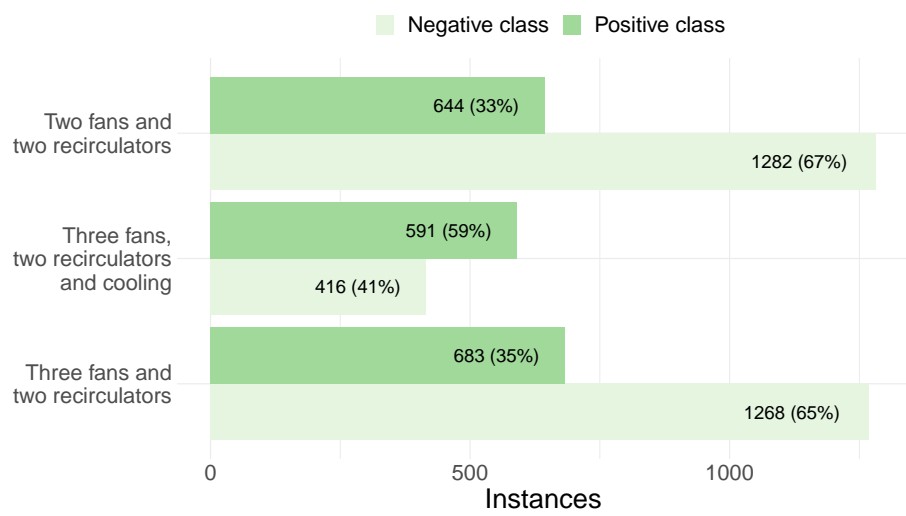

**Figure 2.** Distribution of training classes before manual data augmentation.

Manual data augmentation was applied to the training data to add new observations, from both the positive and negative classes. For the first case, new examples were generated by modifying the on-time of the systems, indicating that, if the target was reached in $S$ seconds with certain internal and external conditions, then, for on-times in the interval $[S + 10, S + 120]$ seconds, the target would also be reached for those same conditions. The number of new examples generated from the positive class depended on the distribution of the classes in the different combinations of weather systems. It was seven new examples for each observation where the target was reached in the case of "two fans and two recirculators" and "three fans and two recirculators", and five new examples for each observation where the target was reached in the case of "three fans, recirculators and cooling". On the other hand, if given certain conditions, where the turn-on of the systems failed to reach the target for a certain time $S$, then, for turn-on times $0 \leq S' < S$ seconds, the targets would not be reached either. To generate new examples, the interval $[0, S - 10]$ was partitioned into

three equal parts and new instances were created, keeping the rest of the conditions and adding the new turn-on times. Finally, another three new examples were generated for each instance of the initial positive class, indicating that, if a temperature $T$ was reached in a given time, then temperatures of $T - 1$, $T - 2$ and $T - 3$ degrees Celsius could not be reached in the same time.

In the case of the two fans and the two recirculators, 2407 observations were available, 1602 (67%) of them failed to lower the temperature inside the greenhouse and 805 were successful (33%). The stratified random split between training/validation and the test (80–20%) provided 1926 observations for the training and validation of the models and 481 for the test. After applying the data augmentation, a training set of 12,140 instances, 6988 of the negative class (58%) and 5152 of the positive class (42%), became available.

For the three fans and the two recirculators, there were 2438 observations, of which, 1585 (65%) of them failed to lower the temperature inside the greenhouse and 853 were successful (35%). The stratified random split between training/validation and the test (80–20%) provided 1951 observations for the training and validation of the models and 487 for the test. After applying the data augmentation, a training set of 12,539 instances, 7075 from the negative class (56%) and 5464 from the positive class (44%), became available.

For the three fans, the two recirculators and the cooling, 1257 observations were available, 519 (41%) of them failed to lower the temperature inside the greenhouse and 738 were successful (59%). When performing the stratified random division between training/validation and the test (80–20%), 1007 observations were available for the training and validation of the models and 250 for the test. After applying the data augmentation, a training set of 6959 instances, 3413 of the negative class (49%) and 3546 of the positive class (51%), became available.

The above process increased the number of instances available to train the models, balanced the distribution of classes and added new unseen examples to increase the generalizability of the models, as can be seen in Figure 3.

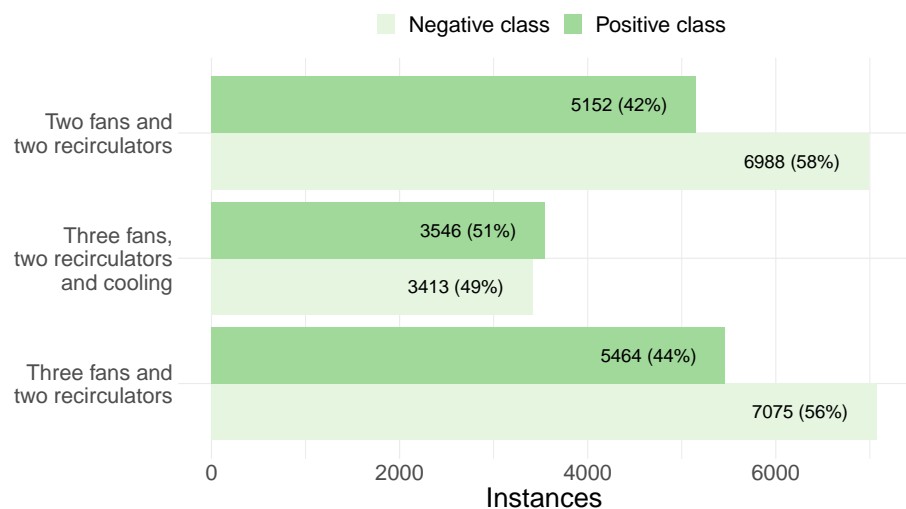

**Figure 3.** Distribution of training classes after manual data augmentation.

## 5. Results and Discussion

### 5.1. Variable Selection

During the experiment, three combinations of climate systems were detected that successfully lowered the temperature inside the greenhouse (the other combinations were not applied or did not lower the temperature). These combinations were: (1) two fans and two recirculators, (2) three fans and two recirculators and (3) three fans, two recirculators and the cooling system. Therefore, we will focus on these combinations of climate systems to estimate whether they will achieve the target temperature and to obtain the final humidity inside the greenhouse that would be reached after switching on for $S$ seconds. For the

analysis at hand, it was firstly important to identify which were the variables that were going to influence the biggest decrease in the temperature of the greenhouse. With this aim, the first task performed was a variable selection using random forests with five-fold cross-validation. Variable selection is necessary because most models do not deal well with a large number of irrelevant variables. These variables will only introduce noise into the model, or, worse, they might lead to over-fitting. Therefore, variable selection serves two purposes: firstly, it helps to determine all of the variables that are related to the outcome, which makes the model complete and accurate. Secondly, it helps in selecting a model with few variables by eliminating irrelevant variables that decrease the precision out of the areas of training and increase the complexity of the model. Ultimately, the variable selection provides a balance between simplicity and fit [31]. This method dropped the variables related to greenhouse internal radiation, wind (speed and direction), rainfall and the use of windows due to the low usage during data collection.

### 5.2. Predictive Models

The initial dataset was divided into three subsets, one for each combination of climate systems. This was carried out to simplify the problem and to try to achieve maximum accuracy. Each subset was divided, by stratified random partitioning, into 80% of the data for the training and validation of the models and the remaining 20% for testing. All data were centred and scaled to have a mean of 0 and a standard deviation of 1. This is important when operating with variables at different scales, as it ensures that they all contribute equally to the result so that no bias is introduced. This ensures that the criteria for finding combinations of predictors is based on how much variation they explain and, therefore, improves numerical stability.

Four classification models were trained: random forest, linear SVM, radial SVM and multiple logistic regression. Ten-fold cross-validation repeated five times was performed to perform an exhaustive hyperparameter search, using 80% of the training/validation data for training and the remaining 20% for model validation. Once the best hyperparameters were obtained, the four models were trained with all of the training data (together with the data generated by data augmentation) and their performance on the test set was evaluated.

All models were trained and evaluated on the same dataset, which makes it possible to compare them and see which one is better. As we can see in Table 3, the random forest model is the most accurate in all cases, significantly improving the results of the other models. If we compare the results obtained with the initial class distribution (see Figure 2), we can see that, in the case of the two fans and the recirculators, the accuracy obtained is improved by 22% if we had classified all of the instances as belonging to the most frequent class (negative class), whereas, for the other two combinations of climatic systems, it is improved by more than 30%. This shows that the achieved models are reasonably good.

**Table 3.** Results of the tests to estimate the humidity inside the greenhouse at each instant.

| Metrics | Radial SVM | Random Forest |
|---|---|---|
| RMSE | 3.55 | 0.16 |
| $R^2$ | 0.96 | 0.99 |
| CVRMSE | 4.42 | 0.20 |
| MAPE | 0.03 | 0.001 |
| MAE | 2.33 | 0.08 |

As all models were trained and evaluated on the same datasets, statistical inference could be performed to determine if there are differences between using one model or another. Using the Student's *t*-test method, the random forest model was the best in all cases and there were significant differences with the other models.

If we analyze in more detail the best model for each combination of climatic systems, we can obtain the following:

- Two fans and two recirculators: The accuracy is 89.2%, as we can see in Figure 4, so the model correctly classifies approximately 90% of the examples. The precision of the model is 92.7%, so it will only be wrong approximately 7% of the time when it predicts that this combination of systems will reach the target. The recall is 90.9%, so the model is able to correctly classify more than 90% of the examples where this combination of systems reaches the target temperature. Regarding the reliability of the model (kappa index), following Landis and Koch's standards [32], we can say that it is substantial and the results are not due to random chance.
- Three fans and two recirculators: The results obtained by this model are very similar to the previous one (see Figure 5).
- Three fans, two recirculators and cooling: This model has an accuracy of 98.4%, as we can see in Figure 6. The accuracy of the model is 100%, so it is absolutely perfect when predicting that this combination of systems will reach the target. The recall is 96.1%, so the model is able to correctly classify more than 96% of the examples where this combination of systems reaches the target temperature. Concerning the reliability of the model, following Landis and Koch's standards [32], we can say that it is almost perfect as the kappa index is very close to 1.

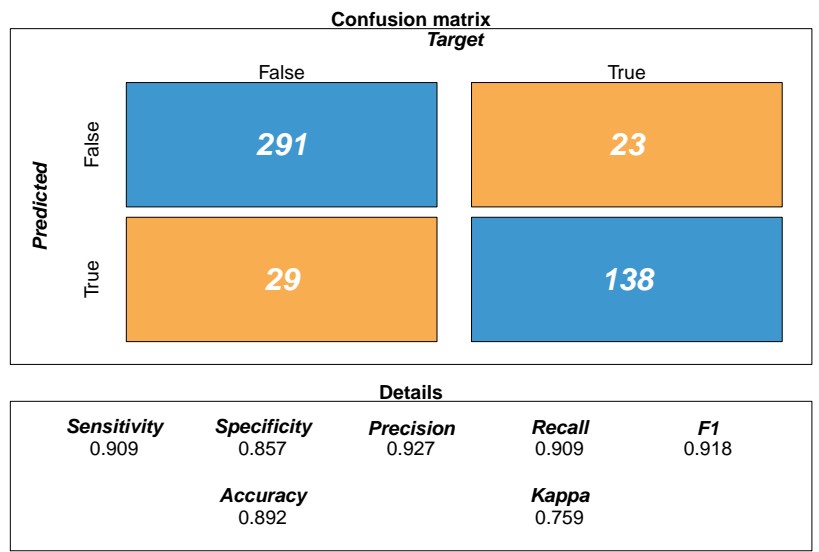

**Figure 4.** Test results of the random forest classification model for two fans and two recirculators.

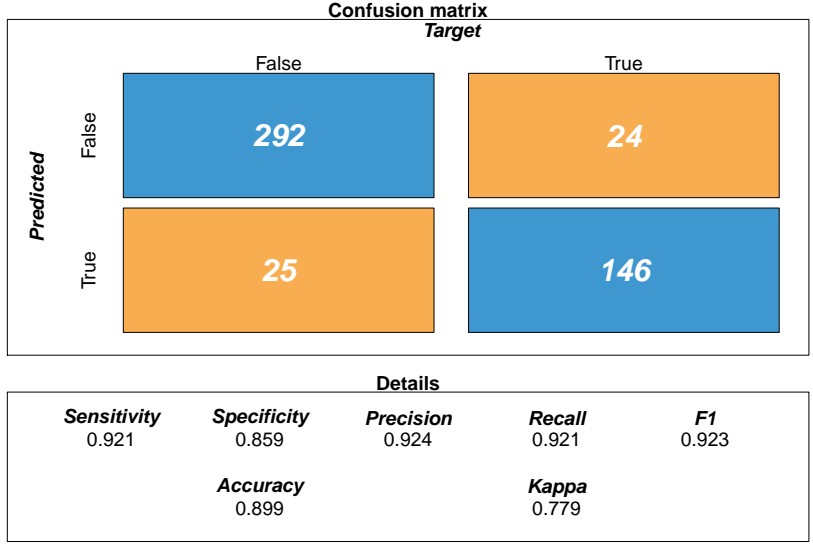

**Figure 5.** Test results of the random forest classification model for three fans and two recirculators.

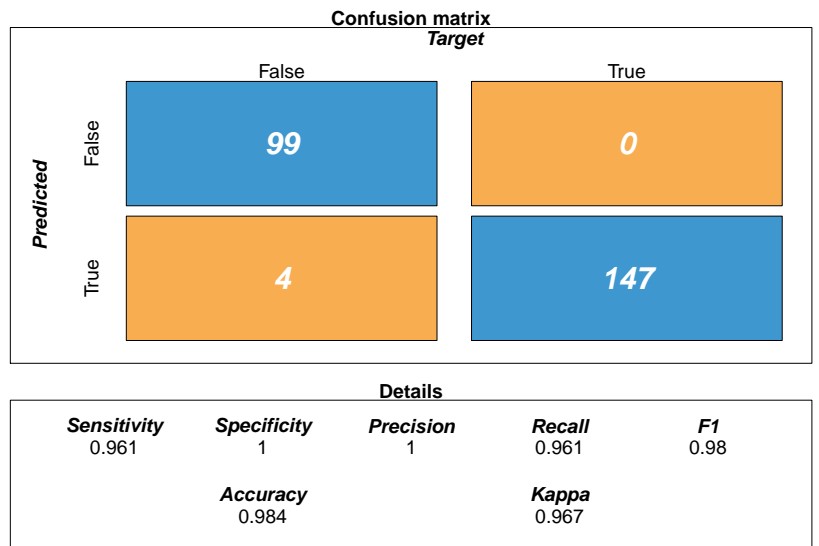

**Figure 6.** Test results of the random forest classification model for three fans, two recirculators and cooling system.

### 5.3. Modeling the Humidity

Another crucial aspect to control in greenhouses is the air humidity since it affects the nutrient uptake [33], water waste and, in general, the quality of the crop. A low humidity results in greater transpiration, water waste and problems in the development of the crops linked, amongst others, to calcium absorption. When conditions are too humid, it may promote the growth of mold and bacteria that cause plants to die and crops to fail, as well as conditions such as root or crown rot. Humid conditions also invite the presence of pests, such as fungus gnats, whose larvae feed on plant roots and thrive in moist soil. Because of this, an algorithm was created to predict the final humidity inside the greenhouse that would be reached when using a certain combination of climatic systems under certain environmental conditions.

For this task, the same steps were followed as in the previous case, but, this time, the four regression models were trained to try to predict the final humidity inside the greenhouse that would occur if the different climate systems were turned on for a certain time. The random forest model performed better in the test dataset than the other models for all evaluation metrics, as can be seen in Table 4. Due to the four models being trained and evaluated on the same datasets, statistical inference can be performed to determine if there are differences between using one or the other. Using the Student's *t*-test method, the differences between the random forest model and the other models were found to be highly significant at 95% confidence. Therefore, the random forest model was chosen to estimate the humidity inside the greenhouse.

**Table 4.** Test results for the estimation of inside final humidity.

| Model | $R^2$ | RMSE | CVRMSE | MAE | MAPE |
|---|---|---|---|---|---|
| Random forest | 0.96 | 1.81 | 2.83 | 1.18 | 0.02 |
| Radial SVM | 0.94 | 2.20 | 3.45 | 1.50 | 0.02 |
| Linear SVM | 0.89 | 3.02 | 4.72 | 2.31 | 0.04 |
| Log. regressor | 0.89 | 2.99 | 4.68 | 2.32 | 0.04 |

In Figure 7, we can see the differences between the predictions of the four models and the expected value in the first twenty observations of the test set.

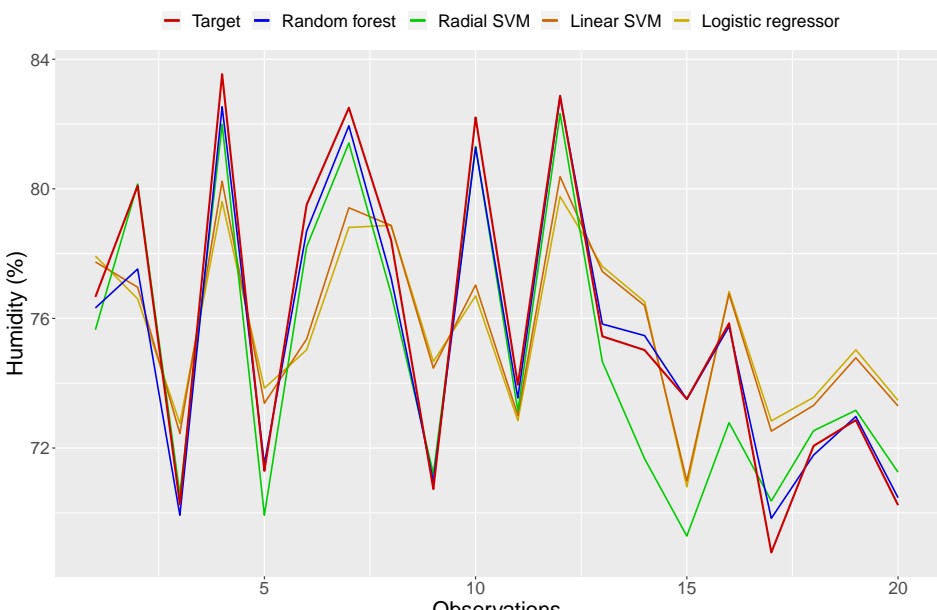

**Figure 7.** Estimation of the final greenhouse indoor humidity for the first twenty observations of the test set using the four regression models together with the true value.

Considering that the final indoor humidity varies between 41.5% and 85% in the test dataset, and the complexity of the problem, an error of 2.83% could be considered as acceptable. These results allow us to predict whether, given current climatic conditions and future predictions for a given time, a combination of climatic systems will be able to achieve the objective or not, at least with a 90% efficiency, which will allow for the automation of the process and the saving of energy and time in which the crops are subjected to undesired conditions, which will increase the yield of the crops. This is a major improvement on the primitive techniques previously used, which were based on trial and error methods that added new systems if the desired conditions were not achieved after a certain period of time. Therefore, when the climatic conditions of the greenhouse are not fulfilled, we can use the three models to decide which of them would reach the target temperature in the shortest time and directly use that combination of climate systems to reduce energy consumption.

### 5.4. Not Observed Cases/Extreme Cases

Evaluating the performance of the models, it was observed that they were able to predict examples similar to the ones that they were trained on reasonably well. However, when testing new examples with "extreme" conditions or conditions that were not similar to the previous ones, it was observed that the models no longer performed as well as expected. For example, when the initial and final temperatures were similar, all of the models predicted that the target could be reached, regardless of the on-time of the systems (even when the on-time was zero). It was also observed that, if a combination of systems was able to reach a temperature $T$ in $S$ seconds when a number of seconds $S'$ (with $S' > S$) outside the usual range of on-time that they had been trained on was introduced, they predicted that the target could not be reached, which was erroneous. Due to these observations, the data augmentation described in Section 4.4 was performed, with the aim of adding new examples and correcting such erroneous predictions.

Figure 8 shows an example of an erroneous prediction of the three fans, recirculators and cooling model, trained without data augmentation, versus the same model trained with data augmentation.

```
> getPred.rf.3FansRecirculatorsCoolingNoAugmented(27, 25, 70, 22, 21, 65, 60, 414, 477, 0)
  Target  Humidity
1 TRUE         70
> getPred.rf.3FansRecirculatorsCooling(27, 25, 70, 22, 21, 65, 60, 414, 477, 0)
  Target  Humidity
1 FALSE        70
```

**Figure 8.** Example of misclassification of a model trained without data augmentation together with the prediction of the same model trained with data augmentation.

### 5.5. Optimized Operation: Machine Learning as a Service

The algorithms here presented can be of help for greenhouse management, especially for the optimization of the yield and for the prevention of catastrophic effects of extreme events such as heat waves and cold snaps. To make sure that the algorithms have an impact and that they can be used by the majority of professionals, providing them as a cloud service that can be accessed by all types of agricultural management solutions was considered. The implementation of them on the cloud was under the use of OpenCPU. OpenCPU is a framework for embedded scientific computing and reproducible research. The OpenCPU server provides a reliable and interoperable HTTP API for data analysis based on R. We used OpenCPU so that our methodology can be integrated into any platform, since all-state in OpenCPU is managed by controlling objects in sessions on a server [34]. For this purpose, an R package was constructed that performs the classification and prediction processes and supports a set of URLs in which the data are stored. The package's functions are the following:

- getPred.rf.2FansRecirculators();
- getPred.rf.3FansRecirculators();
- getPred.rf.3FansRecirculatorsCooling().

All of the package's functions use the following parameters:

- innerTemp0: Initial indoor temperature.
- innerTempF: Final indoor temperature.
- innerHum0: Initial indoor humidity.
- extTemp0: Initial external temperature.
- extTempF: Final external temperature.
- extHum0: Initial external humidity.
- extHumF: Final external humidity.
- extRad0: Initial external radiation.
- extRadF: Final external radiation.
- secondsON: Expected seconds of operation.

In order to access the functionality, we can use OpenCPU in two ways: graphically and by means of a request using curl. Figure 9 shows the graphic user interface of openCPU when making a request with certain inputs, and Figure 10 shows its output. The output consists of a logical value that tells you if, under the input conditions, the desired temperature will be achieved (TRUE in our case), and the final indoor humidity that will be achieved by the selected combination of climatic systems.

Figure 11 shows the same results from a request using curl. This type of request is integrated into our platform in order to make the results accessible to anyone.

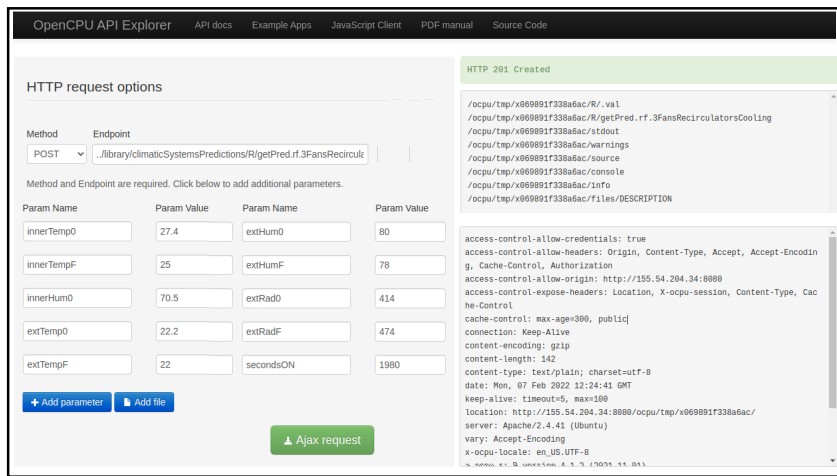

**Figure 9.** Example of a query to the three fans, two recirculators and cooling model to check if it achieves the target under the given conditions using the API.

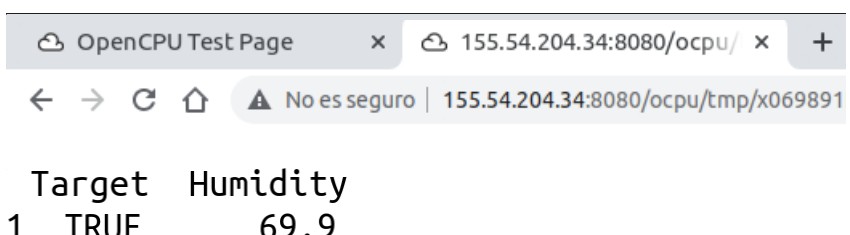

**Figure 10.** Result of query Figure 9.

```
aurora@dibulibu:~/climaticSystemPredictions$ curl http://155.54.204.34:8080/ocpu/library/climaticSystemsPredictions
/R/getPred.rf.3FansRecirculatorsCooling/json?auto_unbox=true -H "Content-Type: application/json" -d '{"innerTemp0":
[27.4], "innerTempF":[25], "innerHum0":[70.5], "extTemp0":[22.2], "extTempF":[22], "extHum0":[80], "extHumF":[78],
"extRad0":[414], "extRadF":[474], "secondsON":[1980]}'
[
  {
    "Target": true,
    "Humidity": 69.9
  }
]
aurora@dibulibu:~/climaticSystemPredictions$ 
```

**Figure 11.** Example of a query to the three fans, two recirculators and cooling model to check if it achieves the target under the given conditions using the terminal.

*5.6. Novelty and Practical Applications*

The research here presented has two facets that make it novel. On the one hand, the methodology used is formed by a comprehensive platform that takes into consideration a large number of data streams, and that offers the possibility of implementing complex AI algorithms. In that respect, we used, for the first time, the support vector machine for the modeling of conditioning systems on greenhouses, which produced outstanding results. On the other hand, we implemented the modeling of temperature and humidity. Although moisture has been seen as a key environmental parameter for the health and yield of crops, humidity had not been included in past studies like the one we present here. We believe that having the model of humidity has made the work highly relevant and realistic for its application in the real world.

Two of the most important environmental parameters that need to be controlled for an optimal greenhouse climate are temperature and relative humidity. Temperature is the most important single parameter in greenhouse controls as temperature has a significant role in plant growth and development. In addition to optimizing the greenhouse temperature, humidity control is of vital importance as optimal plant growth can only be achieved within a certain humidity range.

Modern greenhouses have control systems for these parameters; however, the programming is far from optimizing their operation, saving energy and establishing the optimal set points for crop development. This work contributes significantly to obtaining a

accurate control of temperature and humidity in order to minimize the energy consumption and to establish optimal conditions for plant growth. The results are already being applied to a greenhouse where blueberries are grown commercially.

## 6. Conclusions and Future Work

This paper presents an AI-based mechanism for the smart operation of climatic systems in a greenhouse. We show that we can predict with an accuracy equal to 90% if, under certain conditions, a climatic system will reach a fixed temperature set-point. We are also able to estimate the humidity with a 2.83% CVRMSE. These two machine learning models are crucial for the automatic choice of a combination of the climatic system in order to maintain certain conditions inside a greenhouse and, also, in order to select those that consume less so as to be more efficient. Our mechanism is part of an IoT platform and can be easily integrated into other frameworks since we have developed, through OpenCPU, a package that can be accessed as an API.

For future work, we are currently expanding the climatic systems to be considered. Precisely, we will include those that help to modify the humidity, such as fog machines. In addition, we are analyzing the simplest and more efficient way to lower the temperature in a greenhouse during winter, which is window opening. We are also collecting data from other greenhouses in order to assess how robust our model is with regard to transference, and are testing mechanisms to carry that out.

**Author Contributions:** Conceptualization, A.G.-V., J.M.-B. and A.P.R.; methodology, A.G.-V. and J.M.-B.; software, J.M.-B.; validation, A.G.-V., A.P.R. and M.Á.Z.; formal analysis, A.G.-V.; investigation, A.G.-V., J.M.-B. and V.M.; resources M.Á.Z., V.M. and A.F.S.; data curation, J.M.-B.; writing—original draft preparation, A.G.-V., J.M.-B. and A.P.R.; writing—review and editing, M.Á.Z. and V.M.; visualization, J.M.-B.; supervision, V.M. and A.F.S.; project administration, M.Á.Z. and V.M.; funding acquisition, M.Á.Z., V.M. and A.F.S. All authors have read and agreed to the published version of the manuscript.

**Funding:** This work was sponsored by the European Commission through the H2020 DEMETER (g.a. 857202) project and by PRIMA Foundation and the EU Grant Agreement number: 1821 WA-TERMED4.0 -Call 2018 Section 1 Water. It was also co-financed by the Spanish Ministry of Universities by means of the Margarita Salas linked to the European Union through the NextGenerationEU programme. The publication is also part of the project PERSEO PDC2021-121561-100, funded by MCIN/AEI/10.13039/501100011033 and by the European Union "NextGenerationEU"/"PRTR", and by Berries 4.0: Invernaderos 4.0 para la producción sostenible de superalimentos (2I18SAE00060).

**Institutional Review Board Statement:** Not applicable.

**Informed Consent Statement:** Not applicable.

**Data Availability Statement:** The data presented in this study are available on request from the corresponding author. The data are not publicly available due to privacy restrictions.

**Conflicts of Interest:** The authors declare no conflict of interest.

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
