# Peer review of "Smart Operation of Climatic Systems in a Greenhouse"

_agriculture, doi:10.3390/agriculture12101729_

Round 1
Reviewer 1 Report
This is an interesting work on a timely topic.
Some minor comments:
A. Section 2 is referred to in the introduction section as a review of the literature on AI in greenhouses, which is however not the case. In fact, only the last paragraph of this section disusses on big data and AI application in the context of AI. This could be enhanced with recent state of the art works on AI in PA.
B. There are several typos and grammar mistakes throughout the text. The manusript should be reviewed carefully and revised accordingly.
C. Figures 8-11 could not be read. Please enlarge / improve the quality.
Reviewer 2 Report
The experimental work presented in the Manuscript, entitled „Smart operation of climatic systems in a greenhouse " is interesting research with some promising results. The article reports performed exhaustive data analytics processing that includes the interpolation of missing values, data augmentation and tested several classification and regression algorithms in order to perform the above-mentioned task, there are several shortcomings and modifications that should be included in order to enhance the final manuscript for the readers also the structure of the manuscript must be modified.
Abstract: the authors must rewrite the abstract again. The abstract is poor writing and need to add the important results and the conclusion.
The authors gave more information about the Precision Agriculture in introduction but their works were done under green house. I think, it is better for the authors to write about the smart agriculture. Precision agriculture is the application of remote sensing and GIS under field conditions at large scale. For that, the introduction needs to be written again.
There was a lack of citations in introduction. Please support the introduction by references.
What is the novelty (originality) of the work? And what is new in your work that makes a difference in the body of knowledge? What has been done that goes beyond the existing research.
The author can combine the state of the art in introduction section. Because, the manuscript is complicated for the reader to follow.
Please write the objectives of this study in more details at the end of introduction?
The manuscript from 4. Methodology and data analysis on the prototype to the 6. Optimised operation: machine learning as a service of the manuscript must be separated into two sections, one is materials and methods and other one is results and discussions.
Please, write the practical applications of your work in a separate section, before the conclusions and provide your good perspectives.
Figures 7, 8 and 9 must be improved
Reviewer 3 Report
General comments:
- Over all, the experiment is very well design and the paper well written.
- Additionally, I have some line specific comments below indicating where more detail could be provided.
- More discussion is needed overall. The “results & discussion” section reads almost entirely as just a results section. There is also a general lack of context in relation to other studies – how do your results compare to past studies?
- To help with the discussion, you may find it helpful to reorganize the order of the results & discussion to present HM concentrations in soil followed by EF, CF, and PLI results. Then move on to the HM concentrations in plants and BF results. This may help the story flow a little more logically.
Specific comments:
- Line 19: need a reference here.
- Line 21: need a reference here.
- Line 34: need a reference here.
- Line 37: need a reference here.
- Line 46: Here you say "precision agriculture" but in line 20 you start the sentence with PA. I would prefer to switch in this case to make it more formal.
- Line 81: need a reference here.
- Line 99: need a reference here.
- Line 105: Change "hnks" to thanks. I guess that what you meant.
- Line 123: need a reference here. Mention at least the average high weather temperature for those nine months.
- Line 123: "hottest period" That is not true and misleading. The hottest period in the middle east in general begin to arose between mid-April to the mid May and reach the peak in the months of Jun, July and August then in September start to decline.
- Line 123: " different heights" Explain please the locations of the sencores. Usually, the most be the height of the plants and avoided the direct sun light.
Round 2
Reviewer 2 Report
The authours have improved the manuscript.I have minor comment for section 5.6. Discussion.The title of this section should be changed to the practical applications of this work.
Author Response
We have changed the subsetion's 5.6 title as suggested.
Thank you very much for your revision
Reviewer 3 Report
Thank you very much for your time and efforts taking care of the notes.
Author Response
Thank you very much for your revision